# Role of Diet in Stem and Cancer Stem Cells

**DOI:** 10.3390/ijms23158108

**Published:** 2022-07-23

**Authors:** Francesca Puca, Monica Fedele, Debora Rasio, Sabrina Battista

**Affiliations:** 1Department of Genomic Medicine, The University of Texas MD Anderson Cancer Center, Houston, TX 78705, USA; fra.puca@yahoo.it; 2Department of Oncology, IRBM Science Park SpA, 00071 Pomezia, Italy; 3Institute for Experimental Endocrinology and Oncology (IEOS), National Research Council (CNR), 80131 Naples, Italy; monica.fedele@cnr.it; 4Department of Clinical and Molecular Medicine, La Sapienza University, 00185 Rome, Italy; debora.rasio@gmail.com

**Keywords:** diet, stem cells, cancer stem cells, caloric restriction, asymmetric division, nutrients, autophagy, mTOR, SIRT1

## Abstract

Diet and lifestyle factors greatly affect health and susceptibility to diseases, including cancer. Stem cells’ functions, including their ability to divide asymmetrically, set the rules for tissue homeostasis, contribute to health maintenance, and represent the entry point of cancer occurrence. Stem cell properties result from the complex integration of intrinsic, extrinsic, and systemic factors. In this context, diet-induced metabolic changes can have a profound impact on stem cell fate determination, lineage specification and differentiation. The purpose of this review is to provide a comprehensive description of the multiple “non-metabolic” effects of diet on stem cell functions, including little-known effects such as those on liquid-liquid phase separation and on non-random chromosome segregation (asymmetric division). A deep understanding of the specific dietetic requirements of normal and cancer stem cells may pave the way for the development of nutrition-based targeted therapeutic approaches to improve regenerative and anticancer therapies.

## 1. Introduction

Lifestyle plays a crucial role in health and cancer development. From before conception and during pregnancy, infancy and early childhood, the environment programs an individual’s susceptibility to developing diseases, including tumors, and continues to exert its influence throughout the entire lifespan. Studies on migrant populations support the crucial role of the environment in cancer susceptibility. Cancer rates in migrant populations start shifting from those of the country of origin already after the first decade of migration and will resemble those of the host country within one to two generations [1], highlighting the relevance of changes in dietary and other lifestyle factors in cancer initiation and progression. Forty years have passed since Doll and Peto, with their landmark paper, acknowledged that cancer is a largely avoidable disease and identified smoking and diet as important determinants of health and disease [2]. Two thirds of all cancers can be prevented by refraining from smoking, eating a healthy diet, avoiding excessive weight gain and being physically active [3]. Yet, as studies of molecular pathological epidemiology have highlighted, the exogenous lifestyle factors interact with the individual molecular arrangement, where each neoplasm is the result of a dynamic interplay among environment, host and tumor [4].

Dietary patterns, foods and bioactive food compounds have been found to significantly modify lifespan and health in diverse organisms such as yeast, worms, flies, and mammals [5] and to affect cancer risk and tumor growth, directly or indirectly participating in the cancer process with a profound impact on all cancer’s hallmarks [5,6,7,8,9]. Food quality (what we eat), food quantity (how much we eat) and meal frequency (when we eat) are all aspects that weigh in on the impact of diet on our health system.

Several lines of evidence converge on the idea that stem cells (SCs) are among the major players in orchestrating the response of our body to nutrients, mainly due to their key role in tissue homeostasis. SCs are undifferentiated cells with the potential to generate, upon cell division, different cell types in an organism. They are characterized by unique metabolic features compared to differentiated cells [10,11,12], and they can potentially maintain their undifferentiated state all their life while generating offspring cells committed to differentiation in response to specific needs for tissue homeostasis. In doing so, tissue SCs not only utilize nutrients for their metabolic needs but also adapt their functions, such as self-renewal, autophagy, or differentiation, to the metabolic environment and nutrient availability [12,13,14]. On the other hand, their relatively long lifespan, whereas indispensable to fulfill their function in tissue turnover, holds the back of the coin of continuously being exposed to environmental factors, including diet, and progressively accumulating cell damage at the genetic and epigenetic level, with significant consequences on gene and protein expression and molecular pathways [15,16]. The consequent perturbation in the balance between maintenance and loss of SCs characteristics can be further propagated in the offspring cells, thus inducing alterations in their functionality. In addition, the molecular pathways in charge of sensing nutrient availability also control key SC functions, such as protein synthesis, self-renewal, autophagy, and differentiation, mediating the effects of harmful dietary factors and possibly pushing SCs toward the downhill of cancer transformation, thus generating cancer stem cells (CSCs) [17]. Similarly, some nutrients and dietetic regimens have been studied for their beneficial effects on the regulation of SCs and tissue homeostasis that may result in the inhibition of the cancer transformation process and progression and may prevent the onset of drug resistance [18,19].

Nutrients are normally crucial in SC physiology due to the ability of many nutrient-derived metabolites, released during the catabolic process, to induce chromatin reshaping, epigenetic modifications and gene expression modulation [20]. In this context, for example, glucose has been reported to act as an SC fate regulator, controlling key phases of embryo development [20,21,22]. Similarly, glycolysis and acetyl-CoA have been described to play an active role in the maintenance of pluripotency through induction of histone acetylation; accordingly, modifications of glycolysis or administration of acetyl-CoA precursors can inhibit differentiation and histone deacetylation [11]. Nutrients also serve as donors for moieties involved in post-translational modifications. One example is represented by the hexosamine biosynthetic pathway (HBP) that generates uridine diphosphate GlcNAc (UDP-GlcNAc) for the O-GlcNAcylation of serine and threonine residues embedded in cytoplasmic, nuclear and mitochondrial proteins. In SCs, these post-translational modifications link glucose and nutrient availability with the epigenetic regulation of cell fate determination and differentiation [23]; alterations in this finely regulated process are at the basis of cancer transformation as well as neurodegeneration [24].

Amino acids (AAs) are also involved in SC self-renewal, maintenance of pluripotency and differentiation ability [25]. Several essential AAs (EAAs) are required for the maintenance of both embryonic [26] and adult SCs [27] and their abundance has been demonstrated to increase proliferation, without affecting stemness [21]. In Embryonic Stem Cells (ESCs), threonine and methionine participate in the maintenance of the epigenetic regulation of pluripotency as methyl-group donors for histone methylation. In the absence of methionine or threonine, intracellular S-adenosyl methionine (SAM) levels decrease, DNA and histones methylation is reduced, thus activating p53 expression and inhibiting pluripotency markers and differentiation [25]. Among EAAs, branched-chain amino acids (BCAAs), leucine, isoleucine and valine, represent a major source of acetic acid, hence playing a central role in histone acetylation, epigenetic modifications, transcriptional regulation, and autophagy [5]. BCAAs have been described to have multiple roles in SC physiology and health. Indeed, they have been found to improve satellite cell function [28], Neural Stem Cell (NSC) differentiation and antioxidant defense [29], as well as proliferation and survival of hematopoietic stem cells (HSCs) [25]. Similarly, non-essential AAs (NEAAs) have been shown to be required for skeletal muscle stem and progenitor cell function [30]. Proline, which is abundant in meat and fish, has been shown to preserve the identity of ES cells and to be necessary for their self-renewal and differentiation [31]. Increased proline availability induces mouse ES cells to acquire mesenchymal-like features, such as increased motility and invasiveness (“embryonic-stem-to-mesenchymal like transition”-esMT), and to promote the synthesis of proline-rich proteins, such as collagen.

Fatty acids (FAs) represent another class of nutrient-derived molecules; their importance for SC physiology is demonstrated by the presence of a specific lipidome signature found in certain adult SCs, playing a primary role in the regulation of processes such as quiescence and self-renewal, symmetric-asymmetric division, differentiation, cell-niche interaction and cell fate determination [32]. In SCs, a well-balanced combination of FA synthesis (FAS) and FA oxidation (FAO) is indispensable, and inhibition of one or the other leads to SC exhaustion [32]. In particular, FAO was shown to be necessary for the maintenance of quiescence of HSCs, NSCs, Intestina Stem Cells (ISCs) and skeletal Muscle Stem Cells (MuSCs). Studies of in vitro lipid supplementation or deprivation have demonstrated their effects on SC proliferation and/or differentiation potential [33,34,35], bioenergetics [36] and pluripotency [37].

Another class of nutrients that have been recognized for their importance in SC physiology are phytochemicals. Phytochemicals have been increasingly recognized for their beneficial effects on human health because of their fundamental roles in oxidative stress response, inflammation, cell signaling, cell cycle regulation and many other processes [38]. Vitamin A, which derives from carotenoids, is one of the most important and studied phytochemicals and a potent genetic and epigenetic modulator of SC self-renewal, cell differentiation [39], hematopoiesis [40], and HSC dormancy [41]. Among phytochemicals, polyphenols have been reported to modulate the behavior of different SCs, either directly or indirectly, by regulating the microenvironmental niche. In human Mesenchymal Stem Cells (MSCs), polyphenols suppress hydrogen peroxide-induced oxidative stress [42] and induce osteogenic differentiation [43]. In neural progenitors, they were found to modulate antioxidant and anti-inflammatory pathways by regulating key effectors, such as SIRT1, Wnt, NF-kappa B and Nrf2 [44].

Nutrients also play a pivotal role in CSCs, which heavily depend on the surrounding environment for energy supply. Several studies in the last decades have highlighted specific differences in the metabolite content and in the preferential use of certain metabolic pathways in CSCs, such as aerobic glycolysis, FAO, and ketone body catabolism [18]. Further, it has been demonstrated that CSCs can acquire the ability to synthesize specific AAs [9] as part of their metabolic reprogramming to cope with unfavorable microenvironmental conditions, such as hypoxia, nutrient scarcity and acidosis [18,19], ultimately leading to a nutrient-dependent proliferative advantage. Yet, nutrients can hamper CSCs growth or stemness [45]. Phytochemicals, for example, have been suggested to exert chemopreventive effects, mainly due to their antioxidant and anti-inflammatory effects, to their regulatory role in modifying the epigenome, in interaction with key modulators of oncogenic or oncosuppressive mechanisms, as well as with non-coding RNAs and miRNAs, thus potentially hampering the transformation of SCs into CSCs [38,45,46]. Polyphenols have been suggested to target CSCs by regulating transcription factors, membrane-associated receptor tyrosine kinases (RTKs) and their downstream signaling pathways, lipid rafts, fatty acid metabolism as well as methylation, telomerase and proteasome activity [47].

Therefore, elucidating the mechanisms regulating the response of SCs and CSCs to diet has recently gained much interest as they are crucial elements for health preservation, cancer prevention, prognosis, and therapy.

## 2. Molecular and Cellular Effects of Diet on Stem Cells

The extraordinary potential of SCs, either embryonic or adult, resides in their ability to provide tissues with brand new cells throughout life, given their capacity to divide symmetrically or asymmetrically and lead either to SC self-renewal or differentiation. While the underlying balance of this process is controlled by endogenous mechanisms of development and gene regulation, exogenous signals from the microenvironment, including nutrients, can significantly affect SC specification, differentiation and performance, thus acting on aging and disease. Indeed, nutrients may act directly on SCs or indirectly by regulating the SC niche (non-autonomously). Moreover, nutrients can regulate hormone production, which in turn can influence the behavior of SCs and their niche. In response to these direct and indirect stimuli, SCs activate signaling pathways, reprogram their metabolism and gene expression, converting the dietary input into fate decisions (Figure 1).

SC features regulated by nutrients include symmetric/asymmetric division balance, genome and epigenome integrity, gene expression, metabolism and oxidative status, autophagy, self-renewal, differentiation, and exhaustion. SCs adapt their proliferation to nutrients and growth factors’ availability to undergo cell division when nutrients are sufficient. Mechanistically, this tight balance depends on “master regulators”, such as mTORC1, which can sense nutrients and regulate both metabolism and SC fate [14]. On the other hand, intracellular metabolites, such as acetyl-CoA, regulate both metabolic pathways and epigenetic processes [48], thus connecting diet and metabolism with SC functions. This connection is particularly relevant in fate determination for different types of SCs, as SC self-renewal can be achieved by modifying calories or nutrients [16] and skeletal MuSCs from calorie-restricted mice are more efficient in inducing muscle regeneration than those from ad-libitum-fed mice in transplantation experiments [12]. Similarly, induced pluripotent stem cells (iPSCs) can be reprogrammed by manipulating metabolic pathways [14]; whereas MuSC activation requires the shift from fatty acid oxidation (FAO) to glycolysis [49], the upregulation of glycolytic pathways, as well as the inhibition of mitochondrial activity, may be sufficient to induce stemness features in iPSCs and ESCs [50,51,52]. In vivo, extracellular glucose was found to be capable of modifying gene expression in mouse trophoblast SCs, inducing gene expression changes both before and after differentiation [53]. Notably, crucial stemness master regulators, such as OCT4 and OCT1, have been reported to regulate the expression of glycolytic genes [54,55], thereby strengthening the interdependence between metabolism and stemness maintenance. Intriguingly, even CSC characteristics can be modified by diet, as caloric restriction (CR) has been found to inhibit their cancerogenic and metastatic potential [9].

For both SCs and CSCs, the main molecular node connecting diet and function is represented by the AMPK-mTOR-SIRT1 pathway. When the AMP-activated protein kinase (AMPK) senses low cellular ATP levels, induced by fasting (or exercise), it is phosphorylated by the serine–threonine kinase liver kinase B1 (LKB1) and, in turn, it directly or indirectly modulates enzymes involved in glucose [56] and lipid metabolism [57], as well as the mTOR pathway, thus regulating proteostasis and cell growth [58]. AMPK also targets proteins controlling apoptosis (through direct phosphorylation of p53), cell proliferation (cyclin D1), cell polarity [58], differentiation [59], response to hypoxia (HIF1α) and autophagy [60], hence affecting SC fate [58,61]. Moreover, AMPK increases cellular NAD+, which activates the NAD-dependent histone deacetylase SIRT1, affecting gene expression [62], protein synthesis and SC self-renewal [63]. These molecular events ultimately impact SC “performance” and prevent SC transformation into CSCs. Accordingly, mTOR, AMPK and SIRT1 have been found deregulated in tumors [58,64,65] and in correlation with lifestyle factors lowering ATP, such as CR and exercise, which activate AMPK and are associated with lower cancer risk [6].

Even though these pathways are common to stem and non-stem cells, the impact of diet on SCs may be more dramatic for several reasons. First, SCs have unique metabolic needs, which vary depending on their developmental stage [10,29,51]. The activation of these metabolic pathways is essential for specific SC functions, generating nutrient dependencies more profound than those in differentiated cells. For example, CR has been associated with significantly different outcomes on ISCs and Paneth (non-stem) cells [13]. Similarly, mammary tumor-initiating cells are more sensitive to CR/methionine restriction when grown as mammospheres than in monolayers [66]. In addition, SCs are characterized by lower levels of reactive oxygen species (ROS) compared to their more differentiated counterparts; ROS accumulation and the overall intracellular oxidation state, which are greatly affected by diet and nutrients, are pivotal regulators of the balance between self-renewal and differentiation [67].

Notably, SCs ability to self-renew and to generate progeny throughout the organism’s lifetime implies that their exposure to nutrients and microenvironmental factors is somewhat prolonged, hence amplifying the potential effects of diet.

This evidence highlights the importance of elucidating the molecular relationship between diet and SC properties, to identify potential diet-based strategies to improve overall health and prevent metabolic, degenerative, and neoplastic diseases.

### 2.1. Autophagy and Liquid-Liquid Phase Separation (LLPS)

Autophagy is a conserved homeostatic lysosome-mediated and highly selective self-degradation process that eliminates misfolded or undesired macromolecules and damaged organelles [68]. Further, in stress conditions, including nutrient deprivation, high temperature and exercise, cells digest and recycle self-components to generate energy and building blocks to foster cell survival. The ability to perform autophagy is inextricably linked with aging and health since a progressive impairment of this function, due to reduced autophagy-related proteins and decreased dispatch to lysosomes, is a common denominator of aging tissues and age-associated diseases [68]. Accordingly, its activation through pharmacological treatments or dietetic regimens (such as CR) increases lifespan and health [7,61].

Autophagy is an indispensable process for SCs since it plays a role in maintaining stemness [69] and SC function [70]. In adult and embryonic SCs, the preservation of cellular functions entails the punctual elimination of damaged or detrimental proteins as well as flawed organelles that accumulate with age or pathological conditions. By removing undesired material from the cytoplasm, autophagy not only copes with stress conditions but also fulfills basal physiological needs, contributing to the maintenance of SC functions such as quiescence, self-renewal, activation, metabolism, and differentiation, hindering cellular decline and senescence [69,70,71,72].

In this process, once macromolecules with specific chemical and physical characteristics reach a threshold concentration, they segregate into the cell in a process known as “liquid-liquid phase separation” (LLPS), by undergoing liquid-gel or gel-solid transition [73]. In particular, gel-like aggregates can trigger the generation of autophagosomal membranes, which precisely recognize, envelope and shuttle them to lysosomes in a multistep process that culminates in the generation of autolysosomes [71,73]. Whereas LLPS plays a role in several steps of the autophagic process, including autophagosome assembly, modulation of TORC1 activity and the sorting of proteins for degradation, stress conditions, including nutritional modifications, affect both LLPS and the autophagic process.

mTOR and PKA are the main inhibitors, whereas AMPK and SIRT1 are the main activators of this process [74]. In particular, mTOR and AMPK, being functionally located at the crossroads between sensing nutrient availability, regulating metabolic pathways, and controlling autophagy, are able to master these functions in a concerted way. mTORC1 is a multiprotein complex in which the mTOR serine/threonine kinase interacts, among others, with Raptor [75]. In the presence of abundant AAs or high cellular ATP levels and growth factors, two different small GTPases, Rags (Ras-related GTP-binding proteins) and Rheb (Ras homolog enriched in the brain), promote mTORC1 translocation from the cytoplasm to the lysosomal surface [75]. Once activated, mTOR phosphorylates several substrates, such as eukaryotic translation initiation factor 4E-binding protein 1 (4EBP1) and the ribosomal S6 kinase (S6K1), regulating the synthesis of modulators of cell growth, angiogenesis and tumorigenesis, such as MYC, cyclin D1 and hypoxia-inducible factor 1a (HIF1α) [58]. Concomitantly, activated mTOR phosphorylates the three autophagy modulators, autophagy-related protein 13 (ATG13), Unc-51-like autophagy activating kinase 1 (ULK1) and the Vacuolar protein sorting 3 (VPS34), thus inhibiting their assembly into the autophagy-initiating complexes and preventing autophagosome formation by sequestering TFEB/TFE3 into the cytoplasm [73]. In contrast, nutrient deprivation induces mTOR inactivation and switches cell metabolism towards a catabolic mode, stimulating the activity of the master lysosomal/autophagic transcription factors TFEB and TFE3 [73] and leading to ATG13/ULK1 dephosphorylation/activation and autophagy stimulation (Figure 2).

Similarly, low energy supplies and ATP levels induce AMPK phosphorylation via LKB1, leading to mTOR inhibition [58] (Figure 2). Furthermore, low ATP levels increase NAD+ and stimulate a third autophagy player, the NAD+-dependent deacetylase SIRT1. This enzyme mediates the deacetylation of many effectors, including histones, autophagic Atg regulators [7], the p53 tumor suppressor, the DNA repair factor Ku70 and the FOXO3 transcription factor, hence affecting transcription, autophagy, apoptosis, DNA repair, and activation of longevity genes [76]. FOXO3a also mediates autophagy induced by food deprivation in HSCs but not in their more differentiated progeny [77]. Together with mTORC, SIRT1 fosters the expansion of gut adult SCs during CR [63].

Both mTOR and AMPK are crucial to maintaining SC-specific functions. mTOR ensures HSCs quiescence by lowering ROS [78] and stimulates NSC differentiation [79] and myogenesis [80]. In mouse MSCs, mTOR plays a central role in determining cell fate and lineage-specification between adipogenic and osteogenic commitment as follows: activation of mTOR promotes differentiation into adipocytes through the activation of adipocytic genes [81], whereas mTOR inhibition counteracts MSC aging and maintains MSC osteogenic potential [82]. Excessive alcohol consumption leads to osteopenia and reduced osteogenic differentiation of MSCs in mouse models through systematic activation of mTOR, leading to an increase in peroxisome proliferator-activated receptor γ (PPAR-γ) and a reduction of genes responsible for differentiation, such as runt-related transcription factor 2 (RUNX-2) [83]. In epidermal SCs (EpdSCs), WNT-induced hyperproliferation leads to SC exhaustion and aging via mTOR activation [84]. Similarly, in several adult SCs, including MSCs, MuSCs and ISCs, AMPK not only regulates metabolic pathways but also affects differentiation [56,85,86], and its activation reverts mouse epiblast SCs to naive cells [87].

In several types of adult SCs, including MuSCs and HSCs, autophagy is constitutively active and contributes to the maintenance of SC features and the prevention of senescence. Conversely, impairment of autophagy, due to aging or genetic alterations, induces SC senescence caused by failure of proteostasis, compromised mitochondrial function, oxidative stress [69,70], and increased ROS production, leading to SC exhaustion and aging [88]. In particular, autophagy has been shown to play a crucial role in the maintenance and genetic integrity of Lgr5+ ISCs, responsible for intestinal epithelium repair and integrity, either in physiologic or stress-induced conditions. Unlike all intestinal cells, in Lgr5+ ISCs loss of Atg7 leads to increased oxidative stress, altered interaction with the microbiota and impaired DNA repair. As we will better discuss in the following sections, in the mouse intestine, CR inhibits mTORC1 in the niche of Paneth cells, leading to activation of AMPK/SIRT1 and their signaling cascade in the adjacent ISCs [63]. Noticeably, fasting sustains ISC viability and intestine function in the presence of high doses of chemotherapeutic drugs [89]. This is due to the Atg7-dependent ability of Lgr5+ ISCs to repair irradiation-induced DNA damage more effectively than their more differentiated progenitors. Accordingly, fasting-stimulated autophagy prevents doxorubicin and oxaliplatin-induced DNA damage in ISCs, highlighting the role of diet-induced autophagy in SC performance and chemoprotection [90].

In CSCs from several types of tumors, including glioblastoma and colon, mTOR upregulation has been shown to be responsible for self-renewal and tumorigenicity, not only suggesting that sustained stimulation of mTOR induced by hypercaloric diet might trigger SC transformation but also defining mTOR as a potential target for anticancer therapies [91]. However, in CSCs, autophagy plays a pivotal role in enabling plasticity, viability and proliferation, having both promoting and suppressing effects on their activity and survival [92].

Given the role of autophagy in SC performance and health maintenance, a good deal of attention has been focused on finding dietetic strategies or functional foods able to boost this pathway. At the molecular level, AMPK and SIRT1 stimulation of AKT and mTORC1 inhibition are predicted to favor autophagy. From this perspective, CR, fasting, and a ketogenic diet favor SC autophagy by modulating AMPK and mTOR [7,8,61,93,94]. Conversely, a high-fat diet (HFD) inhibits the AMPK pathway in MuSCs, thus compromising cell activation and muscle regeneration [95]. Similarly, low-protein plant-based diets have been found to reduce IGF-1-mediated activation of the AKT-mTORC1 pathway [96]. Some phytochemicals stimulate autophagy, through SIRT1, AMPK or mTOR pathways in both normal and CSCs [7,61,97]. The activation of SIRT1 by thymoquinone, ferulic acid and melatonin has been demonstrated to act on MSCs and preserve bone mass [98]. Spermidine, a polyamine abundant in the Mediterranean diet and found in whole grains, corn, mushrooms, legumes, soy products, and aged cheese, is able to activate autophagy by acting on the same pathways [99]. Resveratrol, found in berries, grapes, pine nuts, and legumes, activates AMPK and SIRT1 [5,7] and inhibits mTOR [100] in different types of SCs.

The ability of some phytochemicals to target CSC autophagy has made them attractive as potential anticancer treatments. Epigallocatechin gallate (ECGC), quercetin and genistein, among all, have been found to hamper AKT function and hence the mTOR pathway [101]. Moreover, ECGC activates AMPK in human breast CSCs, leading to suppression of mTOR, inhibition of growth and up-regulation of the cyclin-dependent kinase inhibitor p21 [102]. Similarly, curcumin was reported to suppress tumorigenic features in CSCs from glioblastoma [103] and liver cancer [104] by acting on mTOR-dependent Atg activation.

### 2.2. Stem Cell Exhaustion

Several intrinsic factors (DNA damage, altered energy metabolism and mitochondrial function, increased ROS levels caused by spillage of electrons from oxidative phosphorylation, accumulation of misfolded proteins), as well as extrinsic determinants (alterations of the SC niche, modifications of systemic and local factors), contribute to SC functional decline and exhaustion by inducing apoptosis or senescence, which leads to a drop in their self-renewal ability and regenerative potential [105]. In particular, this leads to progenitor cell exhaustion, caused by a decrease in the SC asymmetric division rate, which in turn is responsible for tissue and organismal aging and age-related diseases.

Diet and lifestyle factors can significantly influence both intrinsic and extrinsic factors involved in this process. Dietetic interventions, including CR, were reported to up-regulate metabolic genes, such as glutathione peroxidase, catalase and superoxide dismutase (SOD), and to modulate the expression of genes involved in the mitochondrial function and biogenesis, such as peroxisome proliferator-activated receptor gamma co-activator 1 alpha (PGC1), endothelial nitric oxide synthase (eNOS), SIRT1 and mitochondrial transcription factor A (TFAM), hence counteracting cellular oxidative stress and regulating mitochondrial activity [106]. In addition, nutritional regimens have been demonstrated to modulate autophagy, displaying a protective function against the accumulation of misfolded proteins and molecular (including DNA) damage [105].

In *Drosophila* lymph glands, CR and intermittent fasting have been shown to trigger progenitor cell differentiation, including blood cell progenitors, affecting the cellular immune response and counteracting SC exhaustion [107]. Moreover, in the animal world, from worms to mammals, CR slows down aging, likely by reducing ROS and by inhibiting the mTOR pathway. Of note, the mTOR pathway is associated with cell hyperfunction and accelerated senescence, thus leading to an enhancement of SC exhaustion [108]. Fasting also promotes the FAO and improves the function of ISCs during aging [109]. Olive oil consumption has been associated with beneficial effects on virtually all aging-associated processes [110], including SC exhaustion [111]. These benefits have been attributed in part to its high content of monounsaturated FAs and other highly bioactive components, including phenolic compounds such as hydroxytyrosol, tyrosol, caffeic acid, oleuropein aglycone and oleocanthal [111]. Importantly, it has been shown that oleuropein, a polyphenolic compound found in olive oil and olive leaves, stimulates osteoblastogenesis while inhibiting adipogenesis, by enhancing the osteoblastic phenotype instead of the adipocyte differentiation from MSC progenitors in human bone marrow [112]. Therefore, olive oil consumption has been associated with slower skeletal aging, a complex process in which the continuous recruitment of progenitor cells toward adipogenic differentiation leads to their rapid exhaustion and reduced recruitment into the osteoblastic lineage cells and decreased bone formation.

Similarly, another component of olive oil, the flavonoid epigenin, has been shown to favor osteoblastic differentiation while inhibiting the transition of preadipocytes to adipocytes [113]. Further supporting the beneficial effects of olive oil against skeletal age-related diseases is the study by Liu et al. [114]. The authors show that supplementing olive oil in the diets of ovariectomized rats and women who have undergone artificial menopause prevents bone mineral density decline and osteoporosis. Olive phytochemicals, such as oleuropein and other polyphenols, also displayed protective anti-aging effects on hematopoietic progenitors, providing them a survival advantage and modulating their fate towards asymmetric rather than symmetric divisions [115]. Furthermore, oleic acid induces MSCs to secrete angiogenic factors and acts as an important enhancer of tissue regeneration [116,117]. Another aging-related SC disease involves the endothelial precursor cells (EPCs); they play an important role in the re-endothelization of damaged blood vessels and in the neovascularization of ischemic tissues but are antagonized by angiotensin II, which is involved in the pathogenesis of hypertension when deregulated, causing EPC senescence. Both oleupeurin and oleacein exert a protective effect on angiotensin II-induced EPC senescence [118]. Aside from olive oil, it is important to mention that the whole so-called “Mediterranean diet” is associated with better health conditions during aging [119]. This diet has its origins in Greece, Italy and Spain and is enriched in plant-based foods such as fruits, vegetables, olive oil, legumes, grains, nuts, and seeds. It also includes fish and a moderate intake of red wine around mealtimes, while red meat, high-fat dairy products, and highly processed foods are consumed infrequently. This dietary pattern contains an abundance of bioactive compounds, including a range of vitamins and minerals, polyphenols, fibers, nitrate and mono-unsaturated and polyunsaturated FAs (PUFAs), many of which have been shown, individually or in combination, to elicit beneficial health effects [119]. The Mediterranean diet has been shown to modulate SC exhaustion as well, thus improving EPC fitness and number in elderly people [120,121,122].

### 2.3. Epigenome and Gene Expression

In the last two decades, epigenetic modifications have been associated with SC identity, aging and CSC transformation [123]. Further, nutrition has emerged as a fundamental regulator of the epigenome and gene expression, hence affecting cell metabolism and health [124]. This is achieved by the peculiar capacity of some metabolites to either directly associate with chromatin or indirectly modulate chromatin-modifying enzymes. In SCs, epigenetic modifications of DNA and DNA-associated histones orchestrate their function and fate decisions. Therefore, inputs from the diet can lead to altered chromatin structure and gene expression [26,39,125,126] in embryonic and adult SCs, thus affecting processes such as embryonic development, cell fate determination, cell differentiation, immune function, aging, and oncogenic transformation [127,128], making nutrition, metabolism, epigenetics and SC functions closely correlated to each other [129]. Similarly, in CSCs, some metabolic pathways induce specific epigenetic modifications [10], which discriminate metastatic from primary CSCs and regulate cellular plasticity and aggressiveness [130].

Nutrients introduced by diet are processed into simple metabolites through digestion and, once systemically available, can be uptaken by SCs. These biomolecules can be further catabolized by metabolic enzymes into substrates or cofactors utilized by chromatin-modifying enzymes. In some cases, these enzymes can relocate to the nucleus and catalyze chromatin modifications in the presence of specific cofactors [131]. Nutrient-induced epigenetic modifications may affect both histones (acetylation, acylation, ADP-ribosylation glycation, glycosylation, hydroxylation, methylation, phosphorylation, sumoylation and ubiquitylation) and DNA (methylation and glycation), either enzymatically or non-enzymatically. Metabolites can function as co-factors or substrates for enzymes catalyzing either the addition (“writers”) or the removal (“erasers”) of tagging groups [62]. The newly added chromatin “tag” can induce structural chromatin modifications and phase-separation among differently structured/activated chromatin regions, or it can be recognized and bound by effector proteins (“readers”), modulating gene expression and SC fate (Figure 3). Diet can also modify the expression of epigenetic readers, hence affecting normal SC and CSC function [132].

Methylation at DNA and histone levels is the most well-characterized epigenetic modification. Methionine, together with threonine and metabolites from the one-carbon pathway, such as folate, is a fundamental source of intracellular S-adenosylmethionine (SAM), which acts as a donor of methyl groups for DNA and histones [62]. These modifications modulate the expression of pluripotency genes in adult, fetal SCs and CSCs [133] and regulate mouse ESC maintenance and embryonic development [134]. Accordingly, folate and methionine deficiencies have been found associated with reduced histone methylation, gene expression modifications and neural tube closure (NTC) defects [135]. Conversely, reduction in histone and DNA methylation following methionine restriction, which can be achieved by reducing meat and switching to vegetable-enriched diets, has been shown to affect gene expression and to have beneficial effects on health and longevity [136]. In Triple Negative Breast Cancer (TNBC) CSCs, obesity was found to regulate methylation via induction of an epigenetic reader (methyl-CpG-binding domain protein 2 v2 variant), essential for self-renewal and maintenance of these cells [133], suggesting an additional indirect role of diet on epigenetic information.

Histone acetylation mainly depends on the levels of acetyl-CoA, which is produced from glucose, acetate, ethanol, or FAO [62]. By neutralizing the positive histone charges, this epigenetic modification hampers DNA-histone association and induces chromatin opening and transcription. An increase in glycolysis-derived acetyl-CoA has been shown to be critical to maintaining the pluripotency of human and mouse SCs [137] and promoting their differentiation into regulatory T cells [138].

Histone acylation includes the addition of acyl groups derived from glycolysis (histone lactylation and succinylation), FA and protein metabolism (histone crotonylation), food additives (sodium benzoate-derived histone benzoylation), fasting-, exercise- and ketogenic diet-induced FA and ketogenic amino acid demolition (histone butyrylation and β-hydroxybutyrylation). Histone β-hydroxybutyrate (β-HB) marks active promoters of starvation-responsive genes [139] and is responsible for the protective effect on neurodegeneration, NSC maintenance [93], and small intestine crypt homeostasis [140], by inhibiting histone deacetylases (HDACs) and favoring histone and non-histone acetylation [93].

Histone homocysteinylation (Hcy), which may be driven by increased homocysteine levels in fetal brains, has been shown to modify gene expression and contribute to NTC defects [33]. In SCs, the availability of nutrients, including glucose, also regulates intracellular levels of UDP-GlcNAc, leading to mono-glycosyl-acylation (O-GlcNAcylation) of histones as well as of cellular proteins within the cytosol, mitochondria and nucleus, thus coupling metabolic signals to gene expression, self-renewal and differentiation [141].

Histone ADP-ribosylation requires the addition of mono- or poly-ADP-ribose (PARylation) residues derived from NAD+ to histones. Aging and DNA damage, as well as ROS and HFD, can induce poly(ADP-ribose) polymerase (PARP) enzymes and hence affect chromatin structure and gene expression [62].

Epigenetic modifications of DNA and histones may also take place non-enzymatically, through the addition of acetyl, methyl, and electrophilic groups. The methylglyoxal (MGO) group, derived from increased glycolysis, leads to the formation of advanced glycation end products (AGEs), which are associated with aging, diabetes, and cancer, and are responsible for modifications of nucleosomes, chromatin, and gene expression [142]. Sirtuins mediate the removal of MGO adducts and play an essential role in the maintenance of chromatin integrity. A similar effect can be mediated by ketogenic diet-induced acetoacetate, which inhibits the formation of AGEs by inhibiting MGO synthesis, thus affecting chromatin status and overall health [143].

The intracellular level of certain metabolites can also drive the elimination of epigenetic tags from histones and from DNA. Histone acetylation, for example, is removed by HDACs regulated by metabolites such as β-HB, derived from FAO or ketogenesis. In mouse ISCs, ketone bodies mediated by HDAC1 inhibition increase ISC self-renewal, differentiation, and regenerative capacity [144]. Activation of glycolytic metabolism leads to decreased NAD+ levels, thus reducing the NAD+-dependent histone deacetylase activity of SIRT1, leading to the consequent increase in H4K16 acetylation and satellite cell differentiation [49]. Reduced levels of acetyl-CoA due to HFD have also been associated with the reduction of white adipose tissue in mice [145]. Intracellular alpha-ketoglutarate, which can be derived from glucose and glutamine catabolism, contributes to the maintenance of the ESC pluripotency by promoting histone/DNA demethylation. Similarly, succinate induces differentiation by inhibiting histone demethylases [146]. In squamous cell carcinoma, EpdSCs were found dependent on extracellular serine to maintain elevated H3K27me3 levels, switching to de novo serine synthesis during serine starvation. This metabolic process has been shown to functionally activate α-ketoglutarate-dependent dioxygenases responsible for removing the repressive histone modification and activating cell differentiation and tumor growth inhibition [10].

### 2.4. Symmetric/Asymmetric Division

To maintain tissue homeostasis, tissue SCs undergo asymmetric divisions, leading to the generation of one SC identical to the mother SC and one differentiating cell, which will replace terminally differentiated and dead cells. This mechanism generates highly specialized tissue cells while maintaining a pool of “backup” SCs. However, during tissue growth or repair/regeneration, SCs mainly undergo symmetric divisions to amplify the SC pool. Whereas transient expansion of the SC pool through symmetric division can be physiologic (growth, tissue repair and regeneration), over-activation of symmetric division increases the risk of cancer formation [147]. SC fate and performance are dramatically dependent on a tightly regulated balance between symmetric and asymmetric divisions to keep healthy tissues and prevent cancer onset, progression, and resistance. CSCs, on the other hand, have a reduced ability to divide asymmetrically and mostly undergo symmetric division. Differently from normal SCs, which adapt their division mode to nutrient availability, CSCs continue to proliferate despite nutrient scarcity [148].

It has been demonstrated that restoration of symmetric division may result in an oncosuppressive effect [149]. The maintenance of the balance between symmetric and asymmetric division is controlled by complex machinery that integrates inner and microenvironmental stimuli from the SC niche to finely tune the choice between symmetric and asymmetric division [147].

Asymmetric division is characterized by the biased segregation, before mitosis, of almost all the cell constituents, including organelles, RNA molecules, phosphoinositides, proteins and protein aggregates, cell fate determinants, polarity factors of the cytoskeleton, and centrosomes [32,150,151]. In HSCs, lysosomes, mitophagosomes and autophagosomes are non-randomly distributed, being mostly inherited by the cell retaining SC properties [151]. In this process, genomic material and cell fate determinants require the coordinated segregation into daughter cells with the old DNA and stemness factors co-localizing in the self-renewing undifferentiated daughter cell, and the newly synthesized DNA and cell fate-determining factors directed towards the differentiating cell. This is allowed by the concerted regulation of spindle orientation and cell fate determinant distribution. The Par-3 (Bazooka in *Drosophila*), Par-6 and aPKC form an evolutionarily conserved complex located at the apical part of epithelial and asymmetrically dividing cells, the so-called Par polarity complex, determining cell polarity. The Pins/G-α heterotrimeric protein complex orientates the spindle by recruiting the microtubule-associated protein Numa. Numa is usually located on the lateral side of epithelial cells, leading to a horizontal orientation of the mitotic spindle. Inscuteable (Insc) bridges the Pins/G-α/Numa complex to the Par-polarity complex to generate a microtubule attachment site on the apical cell cortex, leading to an apical-basal mitotic spindle orientation. In the absence of Insc, the Par proteins are not associated with the spindle orientation. Pins/G-α/Numa complex on the lateral side, the mitotic spindle is horizontally oriented, and the cell divides symmetrically [147].

The spindle needs to be correctly oriented with respect to the asymmetric segregation of cell fate determinants. To this aim, once activated by the Aurora A-Par-6 pathway, aPKC, in the Par-polarity complex, phosphorylates the cell fate determinant NUMB, causing a switch in its localization from a uniform peri-cellular distribution to a crescent shape, basal side localization [148].

In addition to organelles, chromosomes are non-randomly distributed to daughter cells as well, in a process known as “non-random chromosome segregation”. During mitosis, the newly synthesized/error-prone DNA is delivered to the differentiating daughter cell, whereas the old DNA is retained in the self-renewing SC [152]. In this process, sister chromatids undergo distinct epigenetic modifications and can be precisely recognized by the spindle microtubules deriving from the old (“mother”) or the new centrosome [153]. The recognition of sister chromatids by microtubules is mediated by centromeres, the chromosome pinch point where the kinetochore is assembled [154]. The key steps in the recognition of old and new chromatids include threonine 3 phosphorylation of pre-existing (old) histone H3 (H3T3P) and asymmetric incorporation of the H3 variant CENP-A, as the previously synthesized protein is segregated in the SC, whereas the newly synthesized is delivered to the differentiating cell [151]. Non-random chromosome segregation represents a powerful mechanism contributing to the preservation of genetic information in the long-lived SC while leaving potential DNA replication errors to the deciduous differentiating cell. By contrast, during symmetric division, DNA molecules are randomly segregated (random chromosome segregation). This implies a casual distribution of replication-associated DNA errors to daughter SCs, promoting cell heterogeneity. The accumulation of oncogenic mutations can promote the cancer transformation process and favor genetic heterogeneity and tumor-associated resistance in established tumors.

Besides the intracellular regulation of asymmetric division, many extrinsic, niche-derived factors (including niche cells, extracellular matrix components and growth factors, such as BMP) can affect the SCs ability to divide asymmetrically. These factors can be lifestyle, diet or age-dependent. Diet can have a dramatic impact on the epigenome by affecting centromeres, preventing non-random chromosome segregation, and compromising the SC ability to divide asymmetrically [155]. Diet and metabolism are likely to influence chromosome segregation and chromosome instability since obesity and metabolic syndromes have been associated with increased micronuclei formation [128].

Growing evidence suggests that the SC division mode can be modulated by diet and nutrient availability. In *Drosophila*, ISCs of the adult midgut adapt their division pattern to nutrient availability, undergoing symmetric division when food is abundant [156], thus leading to a reversible increase in intestinal cell number. Further, the *Drosophila* visual system is characterized by two developmental phases with different sensitivities to dietary inputs [157]. In the early phases of larval development, the number of neural progenitors generated by symmetric division is regulated by nutrients through the insulin/phosphatidyl-inositol-3-kinase (PI3K)/mTOR pathway. Later in the larval stages, neural proliferation becomes diet-independent, and neurogenic asymmetric division is predominant under the ecdysone-mediated action of the Delta/Notch pathway [157]. In conditions of prolonged food scarcity, this strategy preserves neural diversity, and hence visual functionality, at the expense of the total number of neurons [157]. In transplantation experiments, skeletal MuSCs isolated from calorie-restricted mice induce muscle regeneration more efficiently than SCs from ad-libitum-fed mice [12], likely because of nutrient effects on the division mode. Similarly, HFD, high glucose or ad libitum feeding can increase the proliferation of ISCs by stimulating symmetric divisions, leading to a higher frequency of intestinal tumors [158], whereas short-term reduction in food intake or glucose promotes asymmetric divisions [86]. Interestingly, lipids, as well as phosphoinositides, lipid metabolic pathways, lipidic post-transcriptional modification of polarity proteins and interaction of cytoskeletal proteins with lipids, have been found crucial in regulating asymmetric division in several organisms, including mammals [32].

Studies in humans have shown that obese patients have a higher number of adipose-derived SCs [159] with lower differentiation ability than non-obese, metabolically normal individuals [160], suggesting a possible increase in the symmetric/asymmetric division rate in these patients.

Mechanistically, the molecular sensors that respond to nutrient availability and modify metabolic pathways and autophagy also regulate polarity and spindle orientation. The LKB1-AMPK pathway regulates the Par complex activity and epithelial cell polarity [161], hence contributing to the choice of the division mode and raising the incidence of cancer in obese and diabetic patients [58]. The mTORC1 kinase activator SEACAT/GATOR2 complex modulates the mitotic spindle assembly and cytokinesis by regulating the amount of the mitotic kinases Plk and Aurora A, which in turn leads to the localization of mitotic spindle regulators [162]. aPKCζ, which plays a major role in the asymmetric division by phosphorylating and relocating the cell fate determinant NUMB, is activated in low glucose conditions or nutrient deprivation [148]. Accordingly, preliminary data from our laboratory show that glucose restriction increases NUMB phosphorylation in vitro [163], further supporting the crosstalk between diet and asymmetric division.

Intriguingly, aPKCζ is known to regulate glutamine metabolism, and the loss of PKC-ζ, which is observed in CSCs, promotes glutamine utilization and survival in conditions of glucose deprivation [148]. In agreement, reduction of aPKC-ζ levels in mice and humans correlates with increased intestinal tumorigenesis and worst prognosis [148].

Diet may also affect SC polarity by modifying the SC microenvironment. For example, in mice, diet-induced obesity causes mislocalization of Par3, mitotic spindle misalignment, alteration of mammary epithelial polarity, and expansion of the stem/progenitor cell pool through the overactivation of the PI3K/AKT pathway induced by the adipokine leptin. Intriguingly, the alteration of apical polarity proteins is also found in breast tissue samples from high-leptin obese individuals [164].

Therefore, even though these mechanisms have been selected to favor organism adaptation to environmental changes and food availability, they may represent a potential threat in the context of excessive food consumption typical of industrialized countries. It can be speculated that, in humans, overnutrition may be translated into a significant increase in the symmetric division, ultimately causing a boost in the number of mutations in the sensitive SC compartment. Further studies will be required to elucidate these mechanisms.

## 3. Effects of Specific Diets on Stem Cells

### 3.1. Caloric Restriction (CR)

CR is the major nutritional strategy recognized for its effects on SC function and performance. In mice, CR has been shown to reduce IGF-1 [5] and to promote self-renewal and stemness in a variety of SCs, including ISCs, MuSCs, HSCs, lung SCs, NSCs, hair follicle stem cells and MSCs [12,13,85,165,166,167,168,169] (Figure 4), thus regulating tissue homeostasis.

CR can drive several cellular responses, including microenvironment readaptation (local systemic factors and niche cells), ketones production (ketosis) and sirtuin-mediated autophagy [7,61] as well as the SC functions discussed in Section 3 (epigenome and gene expression, SC exhaustion, and symmetric/asymmetric division), ultimately orchestrating energy metabolism (with the reduction of oxidative stress), mitochondrial function, DNA damage repair, and protein homeostasis [105,170].

Mechanistically, the effect of CR on SCs has been thoroughly investigated in the mouse intestine. Surprisingly, LGR5+ ISCs, which are induced to self-renew by a CR regimen, are indeed unable to sense nutrient/energy availability, which is instead perceived by a specific subset of niche cells (Paneth cells), acting both as receivers and transmitters of energetic signals [13,171]. In these cells, CR inhibits mTORC1 and increases the production of the bone stromal antigen 1 (Bst-1) [13], an ectoenzyme responsible for converting NAD+ to cyclic ADP ribose (cADPR). This process stimulates a molecular cascade in ISCs that starts with Ca++ signaling and leads to the sequential AMPK/SIRT1 activation, deacetylation of S6K1 that promotes its phosphorylation by mTORC1, ultimately activating protein synthesis and ISCs self-renewal [63]. Therefore, even though ISCs cannot sense CR directly, they can respond to the cADPR signaling, which results in mTORC1/SIRT1 activation. Accordingly, SIRT1 activation or NAD supplementation mimics the effect of CR in the gut [63]. This process is responsible for increasing the number of ISCs at the expense of the more differentiated transient amplifying cells [13]. Furthermore, it is fundamental to activate intestine cell functions when nutrients become available again [13] and to regenerate wild-type cells, providing a more efficient competition for the niche occupation that lowers the frequency of possible mutated ISCs and cancer incidence [172]. These data highlight the impact of diet on the mutation rate in our cells and on the selective fitness of mutated cells. Further studies would be required to assess if this SC “insulation” process also pertains to other SC types.

In *Drosophila* midgut SCs, decreased expression of the *Indy* (*I’m Not Dead Yet*) gene, a homolog of the mammalian SLC13A5 coding for plasma membrane citrate transporter, is a condition that mimics CR and results in enhancement of mitochondrial biogenesis, reduction of ROS levels and improvement of ISC homeostasis, thus attenuating the aging process and increasing lifespan [173,174].

In the human brain, CR, as well as intermittent fasting, prevents age-associated neurodegeneration and cognitive decline by inhibiting NSC aging, improving synaptic plasticity and stimulating SC neurogenesis. In this context, mitochondrial activation, the generation of scavengers that counteract oxidative stress, epigenetic modifications, and the activation of SIRT, mTOR and Insulin/Insulin-like growth factor-1 pathways, play a major role [175,176].

In MuSCs from mice, CR was found to stimulate cell-autonomous effects by increasing mitochondrial abundance and function, which promotes oxidative metabolism and non-autonomous effects, such as the increase in metabolic regulators/niche factors that trigger SC self-renewal and their regenerative ability [9] and regulate apoptosis [177]. Furthermore, CR improves the regenerative potential of MuSCs harvested from CR donors and engraftment in CR-recipient mice, possibly because of the CR effect on the regulation of the inflammatory response [12].

Given the pivotal role of SCs in cancer onset and the well-recognized effect of CR on SC self-renewal [16], it is worth mentioning that CR and CR mimetics (CRMs) have been studied in the context of CSC generation [66]. Indeed, it has been demonstrated that CR reduces tumorigenesis in breast tissues and inhibits mammary SC self-renewal, thus reducing progenitor cell number [178]. CR was also found to limit the carcinogenic and metastatic potential of CSCs [9] and to reduce radiation-induced leukemogenesis by decreasing the overall number of HSCs [179]. In pancreatic cancer, CR prevents tumor growth by inhibiting StearoylCoA desaturase-1 (SCD), which is responsible for the conversion of saturated fatty acids (SFAs) to monounsaturated fatty acids (MUFAs) [180]. Furthermore, CR downregulates the IGF-1 pathway, thus affecting CSCs through PI3K/AKT/mTOR/S6kinase-dependent mechanisms [170]. CR can affect CSCs indirectly as well, by affecting gut microbiota, thus improving immunosurveillance [9]. Strikingly, CR is more effective than diet cycles or nutrient restriction on tumor growth due to its dramatic effect on the immune system. Indeed, CR was found to lower the number of tumor-promoting immune cells (CD11b+Gr1+) and to increase tumor-fighting (CD8+ and CD4+) immune cells [6]. Recent findings suggest that CRMs may be effective in inducing pharmacological activation of autophagy that stimulates a stem-like phenotype and cytotoxic activity in tumor-infiltrating lymphocytes (TILs) [181]. However, given the well-known impact of diet on DNA damage, the effects of CR and nutritional interventions can be contingent upon the context and vary depending on the age of initiation and duration of the treatment [5].

### 3.2. Prolonged Fasting, Intermittent Fasting, Fasting Mimicking Diet and Ketogenic Diet

Prolonged fasting (PF) and intermittent fasting (IF) are eating patterns that involve cycles of eating and fasting with intervals between meals of 48–120 and 12–16 h respectively, differing from CR, which is characterized by frequent meals very low in calories. A fasting-mimicking diet (FMD) is a plant-based, caloric-restricted diet containing low protein, low sugar, and high unsaturated fat. A ketogenic diet (KD), on the other hand, is a nutritional plan featuring low carbohydrate, high fat and adequate protein supply [8]. All of them are associated with prolonged lifespan, lower glucose levels and increased systemic ketone bodies (ketosis) [61]. At the cellular level, these dietary regimens stimulate autophagy [61] and stemness [69,70,90], provide protection against DNA damage, and mitigate the side effects of chemotherapy [165]. Fasting and FMD were found to induce regeneration in tissues such as the intestine, muscle and pancreas and to favor SC self-renewal in MSCs, ISCs, HSCs, MuSCs [8] and NSCs [182] (Figure 5).

During evolution, food supply used to be intermittent, and fasting-induced stimulation of neurogenesis in the dentate gyrus has had a selective value, allowing a more effective “hunting behavior” in conditions of nutrient scarcity [176].

During eating and fasting cycles, ISCs and intestinal progenitors undergo amplification induced by elevated FAO and modifications in the gut microbiota [8]. In the bone marrow, PF drastically reduces IGF-1 levels and PKA signaling, leading to HSC self-renewal, stress resistance, lineage-balanced regeneration of the immune system with an enhanced commitment to a lymphoid phenotype [183]. Similarly, KD increases MuSCs proliferation in a model of Duchenne muscular dystrophy (DMD) [184] and stimulates self-renewal and fate decisions in ISC [183]. A similar effect is promoted by exogenous ketogenic supplements on neural progenitors [93].

Despite commonalities, PF, IF and KD have some specific outcomes. Fasting and refeeding, for example, significantly stimulate tissue regeneration and rejuvenation through the activation of SCs or progenitor cells, partly due to switching on and off IGF-1, PKA and mTOR signaling pathways, hence promoting autophagy and cell death [8]. In ISCs, 24 h fasting activates FAO, leading to increased ISC functions, enlargement of the ISC compartment and intestinal regeneration [109]. The refeeding phase, which is absent in CR regimens, is particularly important in the regenerative process, characterized by the replacement of damaged cells with new, SC-derived cells [8]. Surprisingly, whereas pre-existing Lgr5+ ISCs tend to decrease in number, PF increases ISCs through the stimulation of non-Lgr5+ cells that replenish the intestinal epithelium in the refeeding phase [185]. Accordingly, CR does not prevent age-associated HSC decline, which is, on the contrary, a feature of short-term fasting or FMD followed by refeeding periods [8]. On the other hand, PF reduces IGF-1 levels in humans [5] and is effective in leading to the complete exhaustion of glycogen storage, inducing a metabolic rewiring toward lipid and ketone utilization [186].

Ketosis and autophagy play major roles in fasting-driven effects. Ketosis, which can also be induced through the administration of exogenous ketogenic supplements, is responsible for several beneficial effects at tissue and cellular level, including activation of anti-inflammatory and antioxidant systems, reduction of insulin levels, neuroprotection, improved mitochondrial function [93] and inhibition of AGE formation [143], hence delaying age-associated diseases and increasing lifespan. The most represented ketone produced during ketosis is β-HB, which is considered the main driver of the anti-aging effect of KD. β-HB is produced in the liver in conditions of glucose depletion following starvation, carbohydrate deprivation, prolonged exercise [187] or ketogenic amino acid administration (i.e., leucine, lysine, phenylalanine, isoleucine, tryptophan, tyrosine, threonine) [188]. Besides its energetic function, β-HB has a key role in cellular signaling, modulating epigenetic modifications and gene expression [140,187]. In the process of β-hydroxybutyrylation, β-HB can function both as a substrate and as a cofactor, inhibiting HDAC function [62]. As a result, in mouse Lgr5+ ISCs of the small intestine (SI), upon fasting, β-HB preserves acetylated lysine residues on histone and non-histone proteins and contributes to SI crypt homeostasis [140]. In the same cells, HDAC1 inhibition by ketone bodies supports the Notch pathway and improves ISC self-renewal, differentiation, and regenerative capacity [144]. In *Drosophila* ISCs, the administration of β-HB prevents age-associated chromatin modifications, such as centrosome amplification and DNA damage, considered early events in tumorigenesis and senescence, while reducing heterochromatin instability in niche cells [189]. Moreover, β-HB can participate in metabolic and signaling pathways by directly binding and modifying proteins, such as G-protein-coupled receptors (GPCRs), free fatty acid receptor 3 (FFAR3) [187] and hydroxycarboxylic acid receptor 2 (HCAR2), thus modulating AMPK activity [93].

Similar to CR, fasting was found to inhibit tumor growth and mitigate the harmful effects of chemotherapy [8,190], due to the induction of ketosis. In this regard, PF is more effective compared to CR and short fasting since it favors the exhaustion of glycogen storage and the complete shunt to lipid and ketone utilization [186]. Moreover, the association of PF or FMD with chemotherapy results in a dramatic reduction of tumor growth due to the pro-regenerative effect on HSCs, limiting chemotherapy-induced immunosuppression [183] and increased immunosurveillance by cytotoxic CD8 (+) tumor-infiltrating lymphocytes (TILs) [191].

Autophagy, which in turn stimulates the DNA damage response and DNA repair [90], also plays a major role in mediating the effects of fasting on SC chemoprotection. In mice SI, autophagy induced by short-term fasting prolongs SC survival in the presence of high doses of chemotherapeutic agents. In contrast, in fed mice, similar doses induce apoptotic cell death in most of the SC compartments [89]. In CSCs derived from TNBC, FMD inhibits the PKA pathway, whereas in non-CSCs it mainly activates PI3K-AKT, mTOR, and CDK4/6, and thus can be selectively targeted to arrest tumor growth [192].

The efficacy of KD in combination with chemotherapy is still being debated. Treatment of glioma SCs with KD or acetoacetate has been shown to induce apoptosis and reduce stemness [193,194,195]. It is worth mentioning that glioblastoma (GBM) cells can utilize ketone bodies and FAs as energetic substrates, even though recent findings showed that the inhibition of the FAs oxidation machinery can induce catastrophic cell death in GMB SCs because of the mitochondrial accumulation of toxic FAs byproducts [196]. Therefore, to date, it is not fully clear what the expected outcome of the KD would be [197]. Further studies will be required to elucidate the underlying mechanisms of these metabolic vulnerabilities in different tumors.

### 3.3. High Fight Diet (HFD)

Lipids represent more than 35% of total calories in HFD and are elevated in butter, animal fat, oily fish and chocolate, as well as in processed foods; the latter, which is largely chosen by young people because it is cheap and tasty, provides a large number of calories, especially from fat and carbohydrates, predisposes to obesity and is strictly associated with several ailments [198].

The effects of lipids on SC functions in an HFD context have only been partially elucidated and are sometimes controversial. However, in different SC systems, FAs overload is associated with detrimental effects on the fitness of the SC compartment through cell-autonomous or niche-dependent mechanisms [165] (Figure 6), potentially leading to impairment of SC functions [199] or their transformation in CSCs [158].

In the intestine, HFD, as well as dietary FAs and cholesterol, has a mitogenic effect on SCs, promoting proliferation and tumorigenesis as follows [35,158]: the mammalian intestine remodels its composition according to nutrient availability by adjusting the number and activity of both Lgr5+ ISCs and progenitor cells (non-ISCs), which differentiate into the different cell types of the intestine. The Lgr5+ ISCs are located at the base of intestinal crypts, close to the Paneth cell niche, where they can easily interact with chemicals coming from food [109]. In particular, HFD or its predominant FA constituents, directly act on ISCs and non-ISCs, which can acquire SC features, augmenting their number through the activation of peroxisome proliferator-activated receptor delta (PPAR-δ), which in turn raises β-catenin activity. The β-catenin function is required for ISC maintenance and, during the early step of intestinal carcinogenesis, is sufficient to induce the precursor adenomatous that can later evolve into intestinal carcinomas [158]. Moreover, HFD cooperates with dysregulated WNT signaling (APC mutations) to alter intestinal bile acids (BAs) and drives malignant transformation in Lgr5+ ISCs, thus promoting the adenoma-to-carcinoma progression [200].

In EpdSCs, HFD induces transcriptional changes in genes controlling extracellular matrix (ECM) and the PI3K pathway [201], whereas, in NSCs, FAs overload has an inhibitory effect on neurogenesis, mainly due to the impact that HFD-derived inflammatory molecules, such as IKKβ/NFκB, have on NSCs [199] (Figure 6). A similar mechanism is also responsible for a feed-forward loop causing the loss of hypothalamic appetite-repressing pro-opiomelanocortin (POMC) neurons, favoring an HFD-induced over-eating behavior [199]. Interestingly, the anti-neurogenic effect of chronic HFD is opposed to the pro-neurogenic effect of dietary restriction, short-term HFD, or administration of specific types of FAs, such as the Omega-3 PUFA [165].

In skeletal MuSCs, HFD inhibits the AMPK pathway, severely compromising cell activation and muscle regeneration [95]. Furthermore, it increases age-associated chromosomal abnormalities in oocytes [202] and reduces the HSC population by inhibiting the signaling of the raft-associated TGF-β receptor [203], while increasing myelopoiesis [204]. In MSCs, HFD increases IL-1, IL-6, TNF-α and NF-κB levels and reduces PPAR-γ expression, potentially affecting the hematopoietic niche and hematopoiesis [205].

Morinaga and colleagues have recently shown that short- or long-term HFD may lead to different effects on hair follicle stem cells (HFSCs) in hair follicles-mini-epithelial organ models. Indeed, short-term HFD induces epidermal differentiation by inducing the accumulation of ROS via aerobic respiration, thus preserving the HFSC pool. Conversely, long-term HFD is associated with autocrine and/or paracrine IL-1R-induced accumulation of lipid droplets and NF-κB activation, which in turn inhibits the sonic hedgehog (SHH) signaling pathway, causing HFSC aberrant differentiation, depletion and ultimately hair loss [206].

These data suggest that HFD activates molecular cancerogenic pathways in SCs. Indeed, HFD has been shown to increase cancer aggressiveness, induce CSC enrichment and reduce survival in mouse models of GBM [207], as well as spur the metastatic potential of CSCs in oral squamous cell carcinomas [208]. This effect is also induced by palmitic acid, an abundant constituent of the western diet, and is dependent on the expression of the FA receptor CD36 [208].

## 4. Conclusions

Diet has a tremendous impact on health, lifespan, disease, and cancer incidence.

SCs, with their unique abilities to last a lifetime and to provide cells for tissue homeostasis, mediate most of these effects by directly or indirectly responding to the dietetic inputs, hence modifying their DNA and functions. Here we have highlighted how diet and nutrients can, permanently or temporarily, affect SC physiology and performance or favor their conversion into CSCs.

A deep understanding of the specific metabolic and nutritional needs of normal and cancer SCs may pave the way for the development of nutrition-based targeted therapeutic approaches to advance regenerative and anticancer therapies, delay aging, and improve disease-free aging.

## Figures and Tables

**Figure 1 ijms-23-08108-f001:**
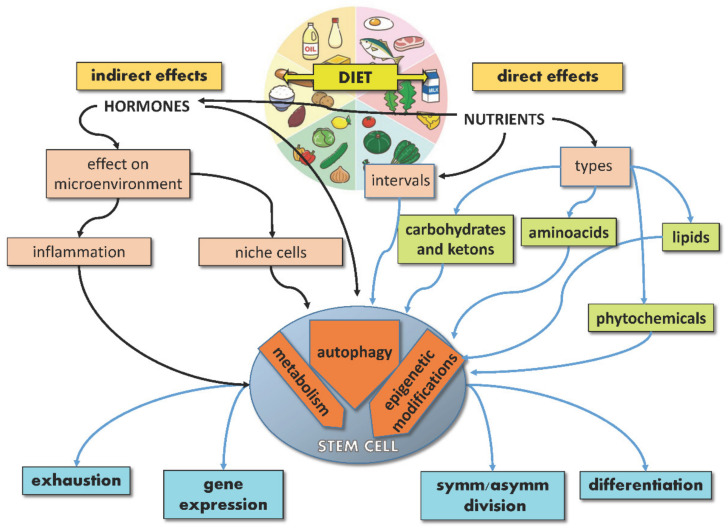
Schematic illustration showing direct and indirect effects of diet on stem cells.

**Figure 2 ijms-23-08108-f002:**
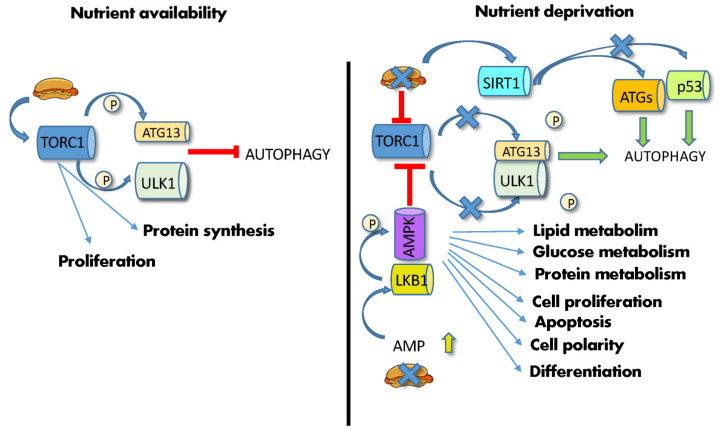
Effects of diet on autophagy. (**Left**) high nutrient availability activates mTORC1, which phosphorylates ATG13 and ULK1, thus inhibiting autophagy. (**Right**) nutrient deprivation and low energy supply leads to increase in AMP and AMPK phosphorylation by LKB1, leading to mTOR inhibition, ATG13/ULK1 dephosphorylation/activation and initiation of autophagy. Low ATP levels also activate SIRT1, which induces the deacetylation of the autophagic ATG proteins, and p53. Abbreviations: AMP: adenosine monophosphate; AMPK: 5′ AMP-activated protein kinase; ATG13: autophagy-related gene; LKB1: liver kinase B1; P: phosphate; p53: tumor protein P53; SIRT1: silent information regulator 1; TORC1: mechanistic target of rapamycin complex; ULK1: Unc-51-like autophagy activating kinase 1.

**Figure 3 ijms-23-08108-f003:**
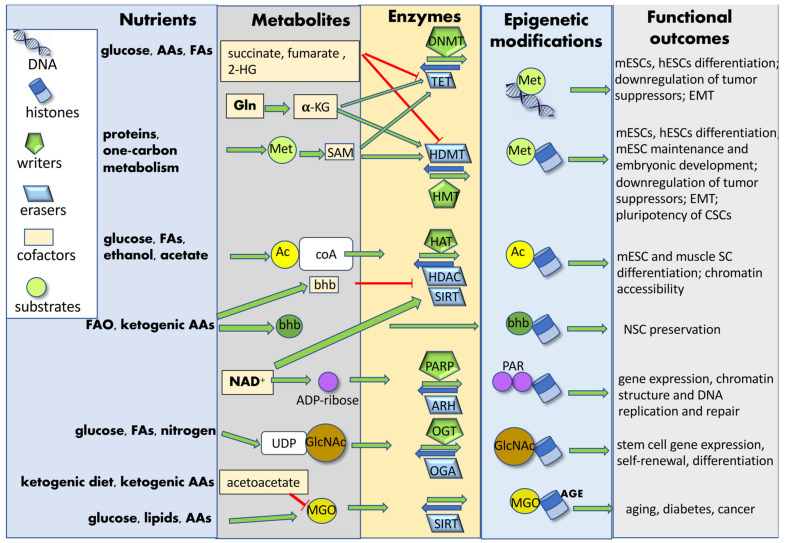
Nutrient-induced epigenetic modifications. Nutrients introduced by diet are processed into simple metabolites and further catabolized by SC metabolic enzymes into substrates or cofactors utilized by chromatin-modifying enzymes. Nutrient-induced epigenetic modifications may affect both histones and DNA, either enzymatically or non-enzymatically. Metabolites can function as co-factors or substrates for enzyme catalyzing either the addition (“writers”) or the removal (“erasers”) of tagging groups. These chromatin modifications finally affect SC gene expression and fate determination. Abbreviations: AAs: amino acids; AGE: advanced glycation end-products; Ac: acetyl group; Ac-CoA: acetyl-coenzyme A; ADP-ribose: adenosine diphosphate ribose; ARH: ADP-ribosyl-hydrolases; bhb: β-hydroxybutyrate; CSCs: cancer stem cells; DNMT: DNA-methyl transferase; EMT: epithelial–mesenchymal transition; FAs: fatty acids; FAO: fatty acid oxidation; GlcNAC: N-acetylglucosamine; Gln: glutamine; HAT: histone acetyltransferase; HDAC: histone deacetylase; HDMT: histone demethylases; hESCs: human embryonic stem cells; 2-HG: 2-hydroxyglutarate; HMT: histone methyltransferase; α-KG: α-ketoglutarate; mESCs: mouse embryonic stem cells; Met: methionine; MGO: methylglyoxal; NAD^+^: nicotinamide-adenine-dinucleotide; NSC: neural stem cell; OGA: O-linked GlcNAc hydrolase; OGT: O-linked GlcNAc transferase; PAR: poly-ADP-ribose group; PARP: poly-ADP-ribose polymerase; SAM: S-adenosyl methionine; SC: stem cell; SIRT: sirtuin; TET: Tet methyl-cytosine dioxygenase; UDP: uridine diphosphate.

**Figure 4 ijms-23-08108-f004:**
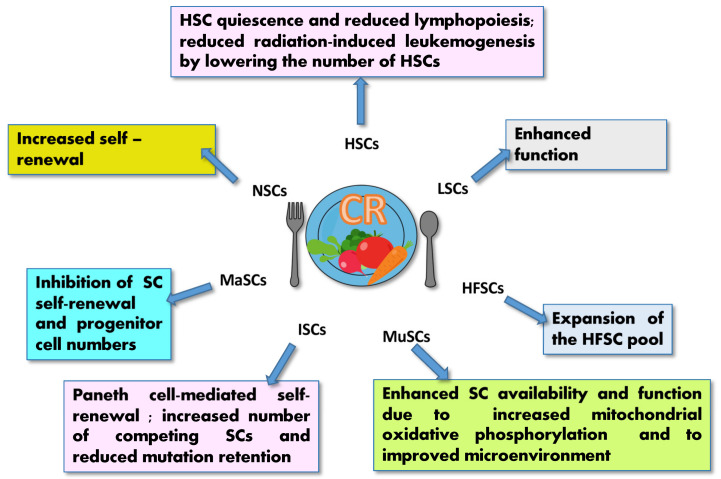
Effects of CR on adult SCs. Abbreviations: HFSCs: hair follicle stem cells; HSCs: hematopoietic stem cells; ISCs: intestinal stem cells; LSCs: lung stem cells; MaSCs: mammary stem cells; MuSCs: muscle stem cells; NSCs: neural stem cells.

**Figure 5 ijms-23-08108-f005:**
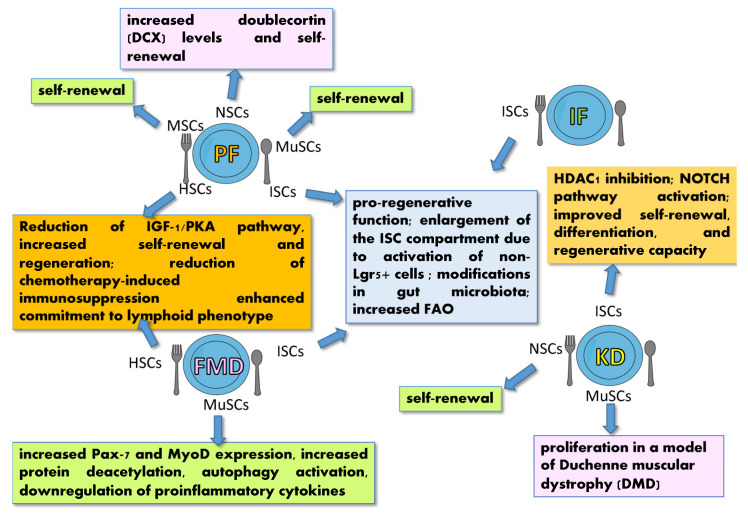
Effects of PF, IF, FMD and KD on adult SCs. Abbreviations: FAO: fatty acid oxidation; HDAC1: histone deacetylase 1; HSCs: hematopoietic stem cells; IGF-1: insulin-like growth factor 1; ISCs: intestinal stem cells; MuSCs: muscle stem cells; NSCs: neural stem cells; PKA: protein kinase A.

**Figure 6 ijms-23-08108-f006:**
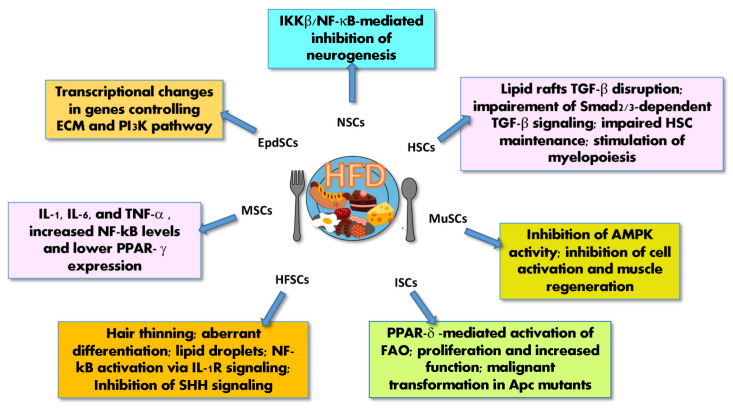
Effects of high fat diet on SCs. Abbreviations: AMPK: 5′ adenosine monophosphate-activated protein kinase; Apc: adenomatous polyposis coli; ECM: extracellular matrix; EpdSCs: epidermal stem cells; HFSCs: hair follicle stem cells; HSCs: hematopoietic stem cells; IKKβ: IkappaB kinase; IL-1: Interleukin-1; IL-6: Interleukin-6; IL-1R: Interleukin-1 receptor; ISCs: intestinal stem cells; MSCs: mesenchymal stem cells; MuSCs: muscle stem cells; NF-κB: nuclear factor kappa B; NSCs: neural stem cells; PI3K: phosphatidyl-inositol-3-kinase; PPAR-δ: peroxisome proliferator-activated receptor δ; PPAR-γ: peroxisome proliferator-activated receptor γ; SHH: sonic hedgehog; Smad2/3: small mother against decapentaplegic 2/3; TGF-β: transforming growth factor β; TNF-α: tumor necrosis factor α.

## Data Availability

Not applicable.

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
