# Peer review of "Role of Diet in Stem and Cancer Stem Cells"

_ijms, 2022, doi:10.3390/ijms23158108_

Round 1

Reviewer 1 Report

The authors wrote a quite interesting review on diet, stem cells and cancer. This is generally of high interest. Topics are hot. This covers good amounts of data although it lacks discussion in some areas.

Diets certainly influence carcinogenic mechanisms. Diet can also interact with other factors, eg, smoking, alcohol, obesity, sleep, exercise, etc. These factors together may influence stem cells, molecular pathology and response to therapy in each patient differentially.

There are also influences of germline genetic variations on diets (appetite and food preference), stem cells and cancer. Gene-by-environment interactions should be discussed.

The authors should discuss such contexts. Research on dietary / lifestyle factors, and personalized molecular biomarkers is needed for cancer outcome research. The authors should discuss molecular pathological epidemiology research that can investigate diet and other factors in relation to molecular pathologies and clinical outcomes. Molecular pathological epidemiology research can be a promising direction and should be discussed, eg, in Ann Rev Pathol 2019; Cancer Causes Cont (published online ahead of print).

Author Response

The authors wrote a quite interesting review on diet, stem cells and cancer. This is generally of high interest. Topics are hot. This covers good amounts of data although it lacks discussion in some areas.

We thank the Reviewer for appreciating our work.

Diets certainly influence carcinogenic mechanisms. Diet can also interact with other factors, eg, smoking, alcohol, obesity, sleep, exercise, etc. These factors together may influence stem cells, molecular pathology and response to therapy in each patient differentially.

 There are also influences of germline genetic variations on diets (appetite and food preference), stem cells and cancer. Gene-by-environment interactions should be discussed.

The authors should discuss such contexts. Research on dietary/lifestyle factors, and personalized molecular biomarkers is needed for cancer outcome research. The authors should discuss molecular pathological epidemiology research that can investigate diet and other factors in relation to molecular pathologies and clinical outcomes. Molecular pathological epidemiology research can be a promising direction and should be discussed, eg, in Ann Rev Pathol 2019; Cancer Causes Cont (published online ahead of print).

We thank the Reviewer for this observation and agree with her/him on the importance of germline genetic variations in diets. Accordingly, in our revised manuscript we have discussed interactions between genes and environmental factors and the importance of a personalized diet based on specific pathologic conditions, referring to the paper by Ogino et al. Insights into Pathogenic Interactions Among Environment, Host, and Tumor at the Crossroads of Molecular Pathology and Epidemiology. Annu Rev Pathol. 2019;14:83-103 (lines 38-41 and ref. 4 of the revised manuscript).

Reviewer 2 Report

The author presented a well-described UpToDate review article describing the role of diet or dietary constituents in stem and cancer stem cells. It is an exciting topic because nowadays, it is known that eating a healthy diet or taking extra supplements or nutraceuticals in daily routine life makes individuals less susceptible to cancer and also helps in anti-aging. However, a few queries need to be addressed before making any decision on this paper.

1.    Did the author focus on any specific kind of cancer SC or its general CSC in the manuscript?

2.    Role of nutrients in normal SC physiology needs to be elaborated more. Many publications need to be added here.

3.    It is good to see if the author can write about the crosstalk or conversion of SC to CSC through dietary intake.

4.    Figure 1 legend must replace ‘Cartoon’ with ‘Schematic Illustration.’

5.    Figure’s font size is minimal and is hardly visible. Increase the font size to an optimum level, making it clear and readable.

Author Response

The author presented a well-described UpToDate review article describing the role of diet or dietary constituents in stem and cancer stem cells. It is an exciting topic because nowadays, it is known that eating a healthy diet or taking extra supplements or nutraceuticals in daily routine life makes individuals less susceptible to cancer and also helps in anti-aging.

We thank the Reviewer for appreciating our work and the impact of the topic.

However, a few queries need to be addressed before making any decision on this paper.

  1. Did the author focus on any specific kind of cancer SC or its general CSC in the manuscript?

We thank the Reviewer for this question. Throughout the manuscript we refer to CSCs in general unless differently specified.

  1. Role of nutrients in normal SC physiology needs to be elaborated more. Many publications need to be added here.

We thank the Reviewer for this suggestion. Accordingly, we have expanded the discussion about the role of nutrients in normal SC physiology, citing several new studies (lines 74-187 and ref. 20-44 of the revised manuscript).

  1. It is good to see if the author can write about the crosstalk or conversion of SC to CSC through dietary intake.

The conversion of SCs to CSCs through dietary intake is indeed a crucial topic in our Review article and it has been discussed several times throughout the manuscript. However, since high fat diet is recognized to be a major stimulator of the conversion of SCs into CSCs, and some of the molecular mechanisms have been identified, we have added a dedicated section on this topic (section 3.3, lines 1407-1471 and ref. 200-210 of the revised manuscript), also including the new Figure 6. Moreover, we have also added a discussion about the role of specific nutrients on preventing the conversion of SC to CSC (lines 196-204 and ref. 45-47 of the revised manuscript).

  1. Figure 1 legend must replace ‘Cartoon’ with ‘Schematic Illustration.’

 We have accordingly modified the Figure legend.

  1. Figure’s font size is minimal and is hardly visible. Increase the font size to an optimum level, making it clear and readable.

We have increased all the Figure’s font size. Thank you for the advice.